# Melatonin Inhibits VEGF-Induced Endothelial Progenitor Cell Angiogenesis in Neovascular Age-Related Macular Degeneration

**DOI:** 10.3390/cells12050799

**Published:** 2023-03-03

**Authors:** Liang-Wei Lin, Shih-Wei Wang, Wei-Chien Huang, Thanh Kieu Huynh, Chao-Yang Lai, Chih-Yuan Ko, Yi-Chin Fong, Jie-Jen Lee, Shun-Fa Yang, Chih-Hsin Tang

**Affiliations:** 1Graduate Institute of Biomedical Sciences, China Medical University, Taichung 403433, Taiwan; 2Department of Medicine, MacKay Medical College, New Taipei City 25245, Taiwan; 3Institute of Biomedical Sciences, Mackay Medical College, New Taipei City 25245, Taiwan; 4School of Pharmacy, College of Pharmacy, Kaohsiung 807378, Taiwan; 5Drug Development Center, China Medical University, Taichung 403433, Taiwan; 6Department of Medical Laboratory Science and Biotechnology, College of Medical and Health Science, Asia University, Taichung 40354, Taiwan; 7Research Center for Cancer Biology and Center for Molecular Medicine, China Medical University, Taichung 403433, Taiwan; 8Center for Molecular Medicine, China Medical University Hospital, Taichung 40402, Taiwan; 9Department of Orthopedic Surgery, China Medical University Hospital, Taichung 40402, Taiwan; 10Department of Sports Medicine, College of Health Care, China Medical University, Taichung 403433, Taiwan; 11Department of Orthopedic Surgery, China Medical University Beigang Hospital, Yun-Lin County 65152, Taiwan; 12Institute of Medicine, Chung Shan Medical University, Taichung 40201, Taiwan; 13Department of Medical Research, Chung Shan Medical University Hospital, Taichung 40201, Taiwan; 14Department of Pharmacology, School of Medicine, China Medical University, Taichung 403433, Taiwan; 15Chinese Medicine Research Center, China Medical University, Taichung 403433, Taiwan; 16Department of Medical Research, China Medical University Hsinchu Hospital, Hsinchu 40402, Taiwan

**Keywords:** melatonin, endothelial progenitor cells, vascular endothelial growth factor (VEGF), platelet-derived growth factor-BB (PDGF-BB), angiogenesis, neovascular age-related macular degeneration

## Abstract

Neovascular age-related macular degeneration (AMD) is described as abnormal angiogenesis in the retina and the leaking of fluid and blood that generates a huge, dark, blind spot in the center of the visual field, causing severe vision loss in over 90% of patients. Bone marrow-derived endothelial progenitor cells (EPCs) contribute to pathologic angiogenesis. Gene expression profiles downloaded from the eyeIntegration v1.0 database for healthy retinas and retinas from patients with neovascular AMD identified significantly higher levels of EPC-specific markers (CD34, CD133) and blood vessel markers (CD31, VEGF) in the neovascular AMD retinas compared with healthy retinas. Melatonin is a hormone that is mainly secreted by the pineal gland, and is also produced in the retina. Whether melatonin affects vascular endothelial growth factor (VEGF)-induced EPC angiogenesis in neovascular AMD is unknown. Our study revealed that melatonin inhibits VEGF-induced stimulation of EPC migration and tube formation. By directly binding with the VEGFR2 extracellular domain, melatonin significantly and dose-dependently inhibited VEGF-induced PDGF-BB expression and angiogenesis in EPCs via c-Src and FAK, NF-κB and AP-1 signaling. The corneal alkali burn model demonstrated that melatonin markedly inhibited EPC angiogenesis and neovascular AMD. Melatonin appears promising for reducing EPC angiogenesis in neovascular AMD.

## 1. Introduction

Age-related macular degeneration (AMD) is an age-related ocular disease that leads to visual impairment, drusen, retinal pigmentary changes, and blood vessel angiogenesis in the retina [1]. Increasing life expectancies and aging populations in most countries worldwide are contributing to a steady increase in the global prevalence of AMD [2]. AMD can be broadly classified as the non-neovascular (dry) or neovascular (wet) type. Neovascular AMD is described as abnormal angiogenesis in the retina and the leaking of fluid and blood that creates a large blind spot in the center of the visual field, causing severe vision loss in more than 90% of patients [3].

Angiogenesis is a complex process that regulates many physiological functions, including wound healing and tissue development, as well as reproduction [4], while it also contributes to pathological processes, such as atherosclerosis [5] and inflammatory diseases [6], as well as neovascular AMD [7]. Angiogenic factors such as vascular endothelial growth factor (VEGF) and platelet-derived growth factor-BB (PDGF-BB) contribute to neovascular AMD [8]. Inhibiting angiogenesis is therefore a critical strategy in neovascular AMD. Currently, anti-VEGF therapy is the only available treatment for neovascular AMD; this therapy targets the vascular endothelial growth factor receptor (VEGFR) [9]. Anti-VEGF/VEGFR2 therapy can effectively inhibit choroidal neovascularization (CNV) and downregulate levels of VEGF mRNA expression in mice with laser-induced CNV [10]. However, although anti-VEGF drugs have provided positive results in clinical trials, these outcomes have not translated into real-world outcomes [3,11]. Thus, other treatment targets besides VEGF and VEGFR are necessary for neovascular AMD.

Endothelial progenitor cells (EPCs) are derived from bone marrow-derived endothelial stem cells, and are involved in physiological and pathological angiogenesis [12]. EPCs are recruited in response to angiogenesis, and regulate several cellular functions such as proliferation and migration [13]. Characterized by their surface markers CD34 and CD133, as well as by VEGFR2, EPCs play an important role in the progression of new blood vessel formation [14]. Importantly, VEGF stimulates EPC angiogenesis, including the survival, motility, and tube formation of EPCs [15,16], via the VEGFR2/c-Src/FAK signaling pathway [17] as well as the transcription factors NF-κB [18] and AP-1 [19]; this mobilization of EPCs enables the development of neovascular AMD [20]. EPC-dependent angiogenesis appears to be a valuable treatment target for neovascular AMD, as the anti-VEGF drug ranibizumab significantly reduces the high levels of circulating EPCs that are involved in AMD angiogenesis [21].

The synthesis and release of the neurohormone melatonin enables organisms to respond to circadian and seasonal rhythms [22]. Melatonin is mainly secreted by the pineal gland under the influence of light stimulation of the retina; to a lesser extent, melatonin is also synthesized within the eye, where melatonin uses ocular structures to mediate a variety of diurnal rhythms and physiological processes within the eye [23]. Increasing the concentration of melatonin at night promotes sleep, while decreasing the concentration of melatonin during the day promotes alertness [23]. Melatonin concentrations tend to decrease with age [24]. The effects of melatonin are beneficial for numerous physiological functions, including the promotion of ocular surface wound healing [25], reductions in inflammation and oxidative stress [24], and angiogenesis [24]. The expression and secretion of VEGF is important in retinal physiology [26]. The release of VEGF from retinal pigment epithelial cells helps to protect neuronal cells and the choroid, maintaining a healthy retina [26]. Notably, melatonin reduces retinal levels of VEGF and protects against ocular angiogenesis diseases [26]. However, whether melatonin affects VEGF-induced EPC angiogenesis in neovascular AMD is unclear. This study therefore aimed to determine whether melatonin treatment inhibits VEGF-induced EPC angiogenesis during the development of neovascular AMD. The study also sought to define any other underlying mechanisms, such as signaling pathways, which may mediate this process.

## 2. Materials and Methods

### 2.1. Materials

Recombinant human VEGF (100-20) was bought from PeproTech (Rocky Hill, NJ, USA). VEGFR2 (07-158), phospho-VEGFR2 (orb106137), CD133 (orb13002), and β-actin (a5441) antibodies, and melatonin (M5250) were purchased from Sigma-Aldrich (St. Louis, MO, USA). Phospho-FAK (3283S) antibodies was purchased from Cell Signaling (Danvers, MA, USA). FAK (sc-1688), phospho-c-Src (sc-12928-R), c-Src (2105S), phospho-c-Jun (sc-822), c-Jun (sc-74543), phospho-p65 (sc-101752), p65 (sc-8008), CD31 (sc-18916), and CD34 (sc-74499) antibodies, as well as the angiotensin II (FAK activator; sc-363643) and c-Src activator (sc-3052), were purchased from Santa Cruz Biotechnology (Dallas, TX, USA). The NF-κB activator (prostratin; ab120880) was purchased from Abcam (Cambridge, MA, USA). PDGF-BB antibody (MBS9404630) was purchased from MyBioSource (San Diego, CA, USA). The VEGFR2 short hairpin RNA (shRNA) plasmid was purchased from the National RNAi Core Facility Platform (Taipei, Taiwan).

### 2.2. Cell Culture

Human primary EPCs were prepared as described in our previous protocols [6]. EPCs were cultured in MV2 complete medium (PromoCell, Heidelberg, Germany) with 20% fetal bovine serum (FBS; HyClone, Logan, UT, USA), and maintained in a humidified incubator at 37 °C, 5% CO_2_ [27].

### 2.3. Analysis of the Eyeintegration v1.0 Database

Gene expression profile records were downloaded from the eyeIntegration v1.0 database (https://eyeIntegration.nei.nih.gov, accessed on 13 January 2022) for 57 normal retina tissue samples and 448 retina tissue samples from AMD patients, for analysis of CD31, CD34, CD133, VEGF, and PDGF-BB expression [28].

### 2.4. Migration Assay

Transwell chambers (8.0 µm pore sizes; Corning, NY, USA) were used for the migration assay. EPCs (8 × 10^3^ in each well) were applied to the upper chambers of 24-well plates in 200 μL of serum-free MV2 complete medium containing 100 ng/mL VEGF. The lower chambers were filled with 300 μL of MV2 complete medium combined with 5% FBS and different concentrations of melatonin (0.1, 0.3, or 1 mM), then incubated for 18 h at 37 °C and 5% CO_2_. The cells in the lower chambers were fixed with 3.7% formaldehyde and stained with 0.5% crystal violet solution. The numbers of migrated cells were counted by microscope (Nikon, Tokyo, Japan) and analyzed by MacBiophotonics ImageJ software [29].

### 2.5. Tube Formation Assay

The 48-well plates were coated with 150 μL of Matrigel (BD Biosciences, MA, USA) and incubated for 30 min at 37 °C. EPCs (2 × 10^4^ cells/well) were seeded onto the gel layer in a 50% MV2 complete medium containing VEGF with melatonin (0.1, 0.3, or 1 mM), then incubated for 6 h at 37 °C. EPC tube formation was evaluated under an inverted phase-contrast microscope (Nikon, Tokyo, Japan). The number of tube branches was quantified by MacBioPhotonics ImageJ software [30].

### 2.6. Proliferation Assay

EPCs (5 × 10^3^) were seeded in a 96-well plate and incubated with melatonin for 24 h or 48 h, before adding 10 μL of CCK-8 reagent (96992, Sigma-Aldrich, St. Louis, MO, USA) and incubating the EPCs at 37 °C for 2–4 h. The samples were quantified by a microplate reader (Bio-Tek, Winooski, VT, USA) at OD 570 nm.

### 2.7. Angiogenesis Protein Array

The cell lysate was examined with the Human Angiogenesis Protein Array (ARY007, R&D Systems, Minneapolis, MN, USA), following the manufacturer’s protocol [31]. The results of the protein array were quantified with the ImageQuant™ LAS 4000 biomolecular imager (GE Healthcare Life Sciences, Chicago, IL, USA).

### 2.8. Chromatin Immunoprecipitation (ChIP) Assay

The chromatin immunoprecipitation analysis followed the previously described methodology [32,33]. DNA was immunoprecipitated with an anti-p65 or anti-c-Jun antibody and purified by phenol-chloroform extraction. Immune complexes were collected with protein G-Sepharose beads and eluted from the beads using an elution buffer. The purified DNA pellet was subjected to PCR, and then the PCR products were analyzed on 1.5% agarose gel electrophoresis and visualized by UV light. The primers 5′-CCAAGAGGCTAGATTCACAGTCAC and 3′-TTCAGCTGTTCCGGCCTTT were used to amplify across the PDGF-BB promoter region (−197/−7).

### 2.9. Chick Chorioallantoic Membrane (CAM) Assay

The CAM assay was performed according to our previously published method [34]. VEGF and melatonin (0.1, 0.3, or 1 mM) were mixed with Matrigel and dropped onto the developing chicken egg, then incubated at 38 °C in an 80% humidified atmosphere. After 14 days, CAMs were observed by microscopy and photographic documentation. The number of CAM blood vessels was quantified by MacBioPhotonics ImageJ software.

### 2.10. Matrigel Plug Assay

The Matrigel plug angiogenesis assay was performed according to our previous research [34,35]. Male nude mice (4 weeks old) were subcutaneously injected with 300 μL Matrigel containing VEGF with the indicated concentrations of melatonin. The plugs were collected after 7 days and processed by immunofluorescence staining for co-staining with CD31, CD34 and CD133 antibodies. All samples were then stained with 40,6-diamidino-2-phenylindole (DAPI) and captured by the TissueFAXS-S-plus imaging system (TissueGnostics, Vienna, Austria). Hemoglobin levels were examined using Drabkin’s reagent and detected at 450 nm with a microplate reader (Bio-Tek, Winooski, VT, USA).

### 2.11. Molecular Docking

The structures of melatonin (PDB code: 5mxb) and VEGFR2 (PDB code: 3V2A) were downloaded from the Protein Data Bank (PDB, https://www.rcsb.org/, accessed on 11 May 2022). The molecular docking analysis between melatonin and the VEGFR2 extracellular domain (ECD) was performed using Discovery Studio software [36].

### 2.12. Corneal Alkali Burn Model

Male C57BL/6J mice (6–8 weeks old) were anesthetized with an intramuscular injection of Zoletil (0.2 mL/kg). Alkali burn injury was induced by placing 2 mm diameter filter paper (soaked in 1 mol/L NaOH) on the center of the right cornea for 30 s, before gently washing the ocular surface with 30 mL of 1X PBS solution. The mice were then immediately treated with melatonin by intraperitoneal (IP) injection (20 mg/kg or 60 mg/kg), or 1X PBS solution by IP injection, or the VEGF inhibitor bevacizumab (5 mg/μL) by eye drop, continuing with the same treatment regimen every 2 days. The eyeballs were observed by a microscope (Nikon, Tokyo, Japan) on days 0, 3, 5, and 7. On day 7, all mice were sacrificed, and their eyeballs were collected and fixed in Davidson’s fluid for histological analysis [37,38]. All animal procedures were performed according to an approved protocol issued by the Institutional Animal Care and Use Committee of China Medical University (Taichung, Taiwan).

### 2.13. Statistical Analysis

All statistical analyses were performed using GraphPad Prism 7.0 software, and all values are expressed as the mean  ±  standard deviation (SD). Differences between selected pairs from the study groups were analyzed for statistical significance using the Student’s *t*-test. One-way analysis of variance (ANOVA) followed by post hoc testing was used for statistical analyses of multiple group comparisons. The difference was considered to be significant if the *p* value was <0.05.

## 3. Results

### 3.1. Higher Levels of EPC Markers Correlate with AMD Progression

Angiogenesis plays an important role in AMD [39]. To confirm the involvement of EPCs in neovascular AMD, we analyzed gene expression profiles from the eyeIntegration v1.0 database. EPC-specific markers CD34 and CD133 and blood vessel markers CD31 and VEGF were all highly expressed in AMD retinas and graded with Minnesota Grading System (MGS) scores of 1 to 4, whereas their expression was low in retinas from healthy individuals (Figure 1), indicating that EPCs contribute to the progression of neovascular AMD.

### 3.2. Melatonin Inhibits VEGF-Promoted Proliferation, Migration, and the Tube Formation of EPCs without Cytotoxicity

In previous research, melatonin treatment has shown strong antiangiogenic effects by suppressing the proliferation, migration and invasion of endothelial cells [40], and melatonin effectively disrupted the tube formation of human umbilical vein endothelial cells (HUVECs) and dose-dependently reduced the viability of HUVECs [41], although the effects of melatonin are unclear in EPCs. When we examined the apoptotic effects of melatonin in EPCs, MTT assay results showed that incubating EPCs with different concentrations of melatonin (0.1–1 mM) for 24 h or 48 h did not affect cell viability (Figure 2A), so this concentration range was used for further experiments. The incubation of EPCs with VEGF (100 ng/mL) and melatonin (0.1–1 mM) for 24 h showed that VEGF alone promoted the proliferation of EPCs; VEGF plus melatonin significantly reduced this proliferation in a concentration-dependent manner (Figure 2B). Next, the migration assay results showed that melatonin significantly inhibited VEGF-induced cell migration in a dose-dependent manner (Figure 2C), while the tube formation assay showed that VEGF alone stimulated the reorganization and the formation of capillary-like structures in EPCs, whereas the addition of melatonin to VEGF significantly suppressed these effects (Figure 2D).

### 3.3. In Vivo Results Show that Melatonin Inhibits VEGF-Induced EPC Recruitment and Angiogenesis

To examine whether melatonin inhibits VEGF-induced angiogenesis in vivo, Matrigel was mixed with VEGF (100 ng/mL) and different concentrations of melatonin as indicated in the CAM assay (Figure 3A). The results showed that melatonin reduced VEGF-induced neoangiogenesis (Figure 3A). Using the Matrigel plug assay to examine the in vivo angiogenesis activity of melatonin, we observed that VEGF promoted microvessel formation and hemoglobin levels over 7 days in the Matrigel plugs, whereas melatonin significantly and dose-dependently inhibited this process (Figure 3B). In the co-immunofluorescent staining results for the blood vessel marker CD31 and EPC-specific markers CD34 and CD133, all markers were significantly and dose-dependently reduced by melatonin (Figure 3C,D). Our findings show that melatonin effectively suppresses VEGF-induced EPC recruitment and angiogenesis in vivo.

### 3.4. Melatonin Inhibits VEGF-Induced Increases in PDGF-BB Expression and EPC Angiogenesis

The results of the angiogenesis protein array show that VEGF treatment increased the amount of angiogenic factors in EPCs and that the addition of melatonin decreased these factors (Figure 4A); in particular, the expression of PDGF-BB was significantly reduced after melatonin treatment (Figure 4B). Furthermore, we found that melatonin decreased VEGF-induced PDGF-BB expression in EPCs in a dose-dependent manner (Figure 4C). Whereas melatonin significantly inhibited VEGF-induced EPC migration and tube formation, these effects were significantly reversed when the EPCs were transfected with PDGF-BB cDNA for 24 h (Figure 4D,E; the tube formation images are shown in Appendix A). Moreover, we found that the levels of *PDGF-BB* gene expression were significantly higher in retinas from patients with neovascular AMD than in retinas from healthy individuals (Figure 4F). These results suggest that melatonin suppresses angiogenesis by inhibiting VEGF-induced PDGF-BB production in EPCs.

### 3.5. Melatonin Reduces VEGF-Induced EPC Angiogenic Functions by Inhibiting the VEGFR2/c-Src/FAK Signaling Pathway

We examined whether melatonin is capable of blocking the VEGF/VEGFR2 interaction. Using molecular docking software to predict melatonin-VEGFR2 binding affinity, we found that melatonin interacts with the VEGFR2 ECD, with an energy value of −27.121 kCal/mol for the docked complex (Figure 5A), indicating that melatonin inhibits VEGF-induced angiogenesis by directly binding with VEGFR2. Next, we found that melatonin inhibits the VEGF-induced phosphorylation of VEGFR2, c-Src, and FAK in EPCs (Figure 5B–D). We then transfected EPCs with VEGFR2 cDNA or treated the EPCs with FAK or c-Src activators to verify whether VEGF/VEGFR2 signaling regulates EPC angiogenesis. Whereas melatonin treatment significantly inhibited VEGF-induced EPC migration and tube formation, these effects were significantly reversed by transfection with VEGFR2 cDNA, and also by treatment with the FAK and c-Src activators (Figure 5E,F; the tube formation images are shown in Appendix A), suggesting that melatonin inhibits the VEGF-induced phosphorylation of VEGFR2, c-Src, and FAK in EPCs via VEGFR2/c-Src/FAK signaling.

### 3.6. NF-κB and AP-1 Contribute to Melatonin-Induced Inhibition of EPC Angiogenesis

VEGF-induced phosphorylation of the NF-κB subunit (p65) and AP-1 subunit (c-Jun) was significantly and dose-dependently reduced by melatonin (Figure 6A,B). Next, we tested whether NF-κB or AP-1 are involved in the melatonin-induced inhibition of EPC angiogenesis. Treatment with NF-κB and AP-1 activators significantly promoted EPC angiogenesis, which was significantly reduced by melatonin treatment (Figure 6C,D). We then observed that melatonin significantly reduced NF-κB or AP-1 luciferase activity and their translocation into the nucleus (Figure 6E,F; the tube formation images are shown in Appendix A). We also explored whether PDGF-BB is involved in the melatonin-induced inhibition of NF-κB or AP-1 transcriptional activation, using the ChIP assay to assess the in vitro recruitment of NF-κB or AP-1 to the PDGF-BB promoter. As shown in Figure 6G,H, VEGF induced the binding of NF-κB or AP-1 to the PDGF-BB element, and melatonin treatment reduced this binding. These results indicate that melatonin inhibits VEGF-induced PDGF-BB production and angiogenesis in EPCs via the suppression of NF-κB and AP-1 activity.

### 3.7. Melatonin Treatment Inhibits EPC Angiogenesis in the Corneal Alkali Burn Model

As shown in Figure 7A, corneal neovascularization was increased in the PBS-treated eyes, and decreased after melatonin treatment in a dose-dependent manner. Notably, the effects of melatonin in the highest dose group (60 mg/kg) were similar to those of the bevacizumab treatment group. After the animals were sacrificed on Day 7, the eyeballs were removed to examine corneal epithelial defects. H&E staining showed that corneal swelling was dose-dependently decreased by melatonin (Figure 7B,C). Co-immunofluorescence staining for EPCs and vessel markers showed that melatonin significantly and dose-dependently inhibited EPC angiogenesis and the recruitment of CD31-, CD34- and CD133-positive colonies (Figure 7D). The effects of melatonin 60 mg/kg were similar to those of bevacizumab treatment. Levels of PDGF-BB expression were increased in the control group and significantly reduced after melatonin or bevacizumab treatment (Figure 7E). Thus, melatonin appears to suppress PDGF-BB production, EPC recruitment, and angiogenesis, as well as AMD development in vivo.

## 4. Discussion

Neovascular AMD is a serious type of late AMD and an angiogenesis-dependent disease [42]. By triggering the mobilization of EPCs from the bone marrow and their recruitment during pathological states, VEGF supports the differentiation of EPCs into mature endothelial cells at angiogenesis sites [43]. EPCs may serve as a biomarker of neovascular AMD, based on the higher levels of EPC expression in blood from patients with AMD compared with blood from healthy controls [21]. Our analysis of gene expression profiles downloaded from the eyeIntegration v1.0 database confirmed high levels of EPC-specific markers (CD34 and CD133) in the retinal tissue of patients with AMD. Our preclinical findings also indicated that EPCs contribute to angiogenesis and AMD progression. Our results are consistent with pre-existing clinical evidence and emphasize the importance of EPCs in neovascular AMD.

VEGF is described as an endothelial cell-specific mitogen [44] that promotes the proliferation of new vessels and increases vascular permeability, and is considered to be a critical angiogenic factor of all angiogenesis processes [45]. Anti-VEGF treatment is an established therapeutic approach for neovascular AMD [46]. However, although the existing anti-VEGF agents, including bevacizumab, effectively inhibit VEGF activation [47] and EPC-derived pathological angiogenesis [48], many patients do not show the expected efficacy of anti-VEGF agents after repeated administration [11].

EPC-targeting therapies may therefore be a promising option to inhibit the angiogenesis process in neovascular AMD. In this study, EPCs were used to investigate the antiangiogenic effects of melatonin. Our findings show that melatonin inhibits VEGF-induced EPC angiogenesis in a concentration-dependent manner, without any evidence of cytotoxicity. Moreover, melatonin significantly suppressed the VEGF-induced stimulation of EPC recruitment and angiogenesis in vivo. Importantly, the corneal alkali burn model indicated that the inhibitory effects of melatonin on EPC-derived pathological angiogenesis are similar to those of bevacizumab, indicating that melatonin has potential against EPC-derived angiogenesis in neovascular AMD.

There is much evidence to indicate that the binding of VEGF to VEGFR2 regulates angiogenesis progression in physiological and pathological conditions, and promotes EPC-derived angiogenesis [49,50]. In our study, we found that melatonin inhibits VEGF-induced angiogenesis in EPCs by blocking the VEGF/VEGFR2 signaling pathway. Notably, molecular docking software results identified a good affinity between melatonin and the VEGFR2 ECD. To the best of our knowledge, this study is the first to identify that melatonin has the potential to bind with VEGFR2 and that melatonin may compete with VEGF in this binding process. Our in vitro evidence also showed that melatonin dramatically reduced the phosphorylation of VEGFR2 in EPCs. All of these results reveal that melatonin inhibits EPC-derived angiogenesis by regulating VEGF/VEGFR2 activity.

VEGF and PDGF-BB, as well as their receptors, are critical for normal and pathologic angiogenesis [51]. In neovascular AMD, VEGF promotes new vessel growth, while PDGF-BB maintains the interaction between pericytes and endothelial cells in maturing vessels [51]. While VEGF antagonists can inhibit angiogenesis in neovascular AMD, dual VEGF/PDGF inhibitors have proven to be even more beneficial in inhibiting angiogenesis in human endothelial cells and human pericytes in neovascular AMD [51], as well as in suppressing neovascularization in a mouse model of laser-induced CNV [52]. In our study, melatonin significantly reduced VEGF-induced PDGF-BB production in EPCs, as well as EPC migration and tube formation. In addition, our findings from the corneal alkali burn model suggest that melatonin can significantly inhibit corneal levels of PDGF-BB expression and angiogenic activity. Our results suggest that melatonin suppresses angiogenesis in EPCs by inhibiting PDGF-BB production.

The promotion or inhibition of angiogenesis is part of the homeostatic balance, with positive and negative effects outside the optimum range. Melatonin influences this balance, with evidence from several clinical research investigations demonstrating that this hormone has antiangiogenic effects in cancer and chronic ocular diseases. For instance, in cancer treatment, adjuvant melatonin appears to be very effective in early-stage disease and helps to reduce the side effect profiles after radiotherapy and chemotherapy [53]. In patients with central serous chorioretinopathy, treatment with oral melatonin (3 mg) three times daily for 1 month resulted in significant improvements from baseline in best-corrected visual acuity (BCVA) and decreases in central macular thickness (CMT), without any adverse effects [54]. The attractive side effects profile and relatively low cost of melatonin suggest that this hormone may be appropriate in a chronic ocular disease such as AMD, although supporting data from large prospective studies are needed before melatonin can be used in the clinic [54]. The FDA-approved the anti-VEGF agents bevacizumab, aflibercept, and ranibizumab, and have shown good therapeutic results in neovascular AMD, but anti-VEGF agents are limited by the necessity for monthly injections in the clinic and the long-term nature of treatment [55]. Moreover, the high cost of anti-VEGF therapy is a heavy burden for patients [56]. Our study evidence suggests that melatonin may overcome new existing therapeutic obstacles with anti-VEGF agents and offer a novel option for treating neovascular AMD.

In conclusion, our study indicates that melatonin inhibits VEGF-induced increases in PDGF-BB expression in EPCs by inhibiting the signaling of VEGFR2, c-Src, FAK, NF-κB and AP-1, all of which appear to effectively inhibit EPC angiogenesis (Figure 8). Thus, melatonin shows promising therapeutic potential, alone and in combination with a VEGF inhibitor, for neovascular AMD.

## Figures and Tables

**Figure 1 cells-12-00799-f001:**
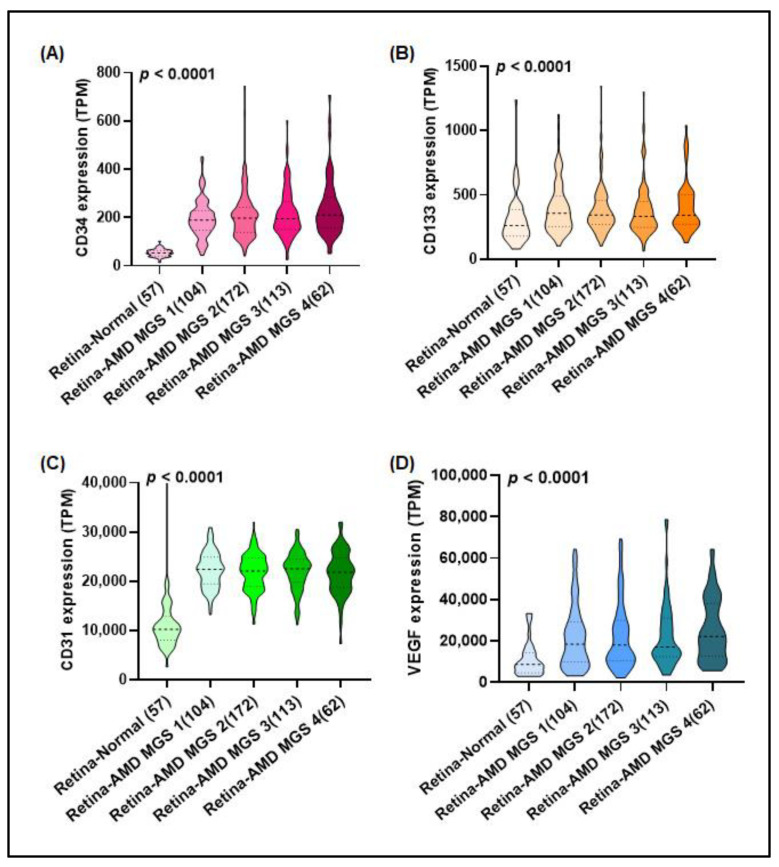
(**A**–**D**) High levels of EPCs and blood vessel markers in AMD retinas were graded by Minnesota Grading System (MGS) scores. Levels of EPC-specific markers (CD34 and CD133) and blood vessel markers (CD31 and VEGF) were analyzed in gene expression profiles of normal retina tissue and neovascular AMD tissue samples downloaded from the eyeIntegration v1.0 database. *p* < 0.05 compared with the normal group.

**Figure 2 cells-12-00799-f002:**
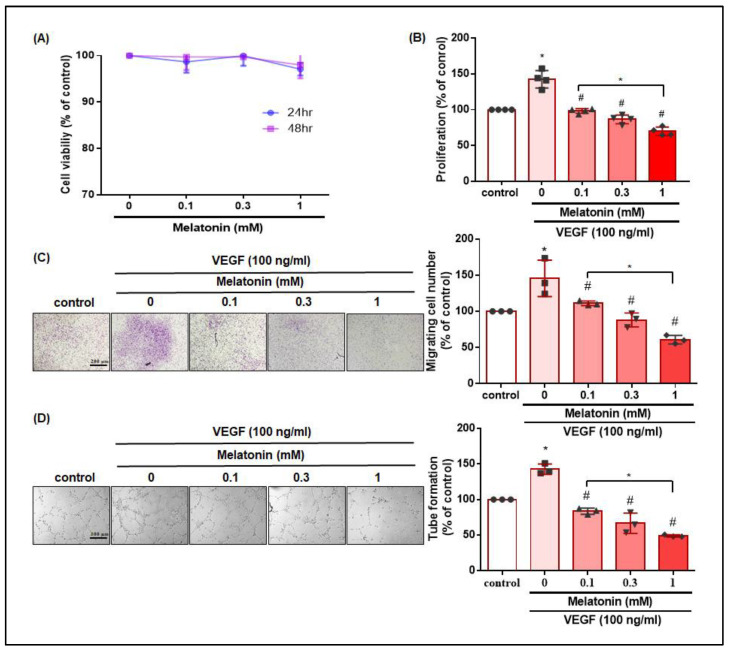
Melatonin decreases VEGF-induced EPC proliferation, cell migration, and tube formation without cytotoxic effects. (**A**) EPCs were incubated with melatonin (0.1–1 mM) for 24 h or 48 h, and cell viability was examined using the MTT assay (*n* = 4). (**B**) EPCs were incubated with VEGF (100 ng/mL) and melatonin (0.1–1 mM) for 24 h. Cell proliferation (*n* = 4) was examined by the CCK-8 assay. (**C**) EPCs were incubated with VEGF (100 ng/mL), and different concentrations of melatonin (0.1–1 mM) for 18 h. Cell migration (*n* = 3) was examined by the Transwell assay. (**D**) EPCs were incubated with VEGF (100 ng/mL) and different concentrations of melatonin (0.1–1 mM) for 6 h. The capillary-like structure formation was determined by the tube formation assay (*n* = 3). * *p* < 0.05 compared with the control group; # *p* < 0.05 compared with the VEGF-treated group.

**Figure 3 cells-12-00799-f003:**
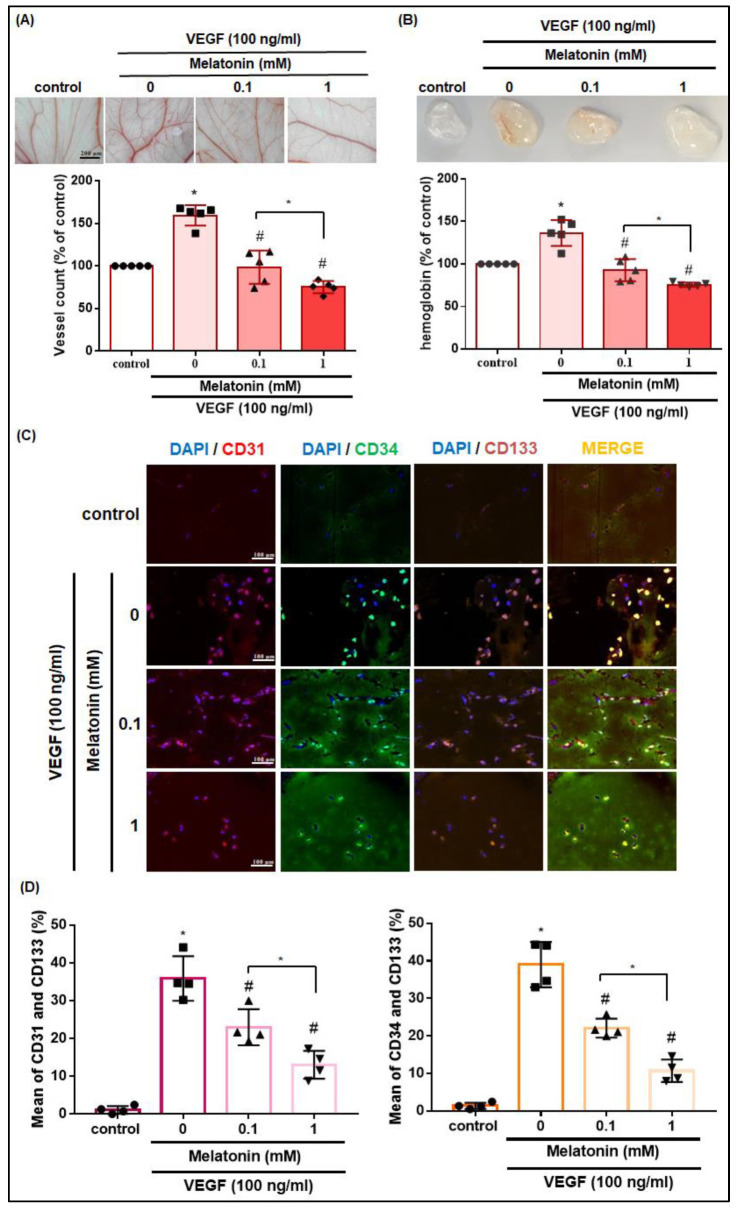
Effects of melatonin on VEGF-induced EPC angiogenesis in vivo. (**A**) Five-day-old fertilized chick embryos (*n* = 5) were treated with VEGF (100 ng/mL) and different concentrations of melatonin (0.1–1 mM) for 14 days. After treatment, the CAMs were examined by microscopy and photographed. (**B**–**D**) Matrigel plugs were treated with PBS (control group) or VEGF (100 ng/mL) with different concentrations of melatonin (0.1–1 mM) and subcutaneously injected into the flanks of nude mice (*n* = 5). After 7 days, the plugs were photographed, and then hemoglobin levels were quantified and visualized by co-immunofluorescence staining at 20× magnification for CD31, CD34, and CD133 antibodies. * *p* < 0.05 compared with the control group; # *p* < 0.05 compared with the VEGF-treated group.

**Figure 4 cells-12-00799-f004:**
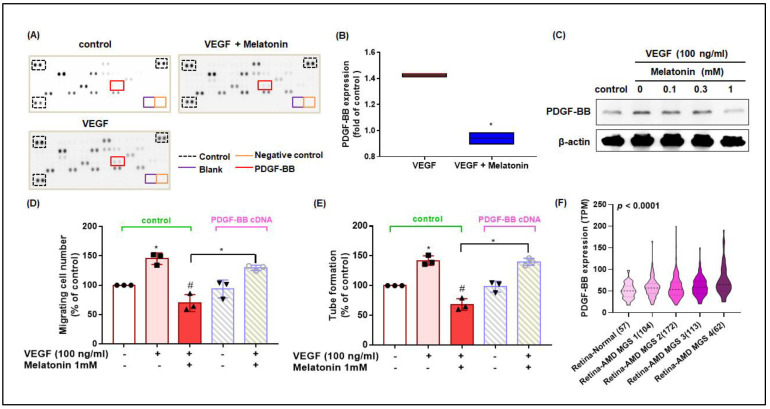
Melatonin reduces VEGF-induced EPC migration and tube formation by inhibiting PDGF-BB production. (**A**) EPCs were incubated with VEGF (100 ng/mL) alone and in combination with melatonin (1 mM) for 24 h. Cell lysates were collected from each treatment condition and from untreated EPCs, and then profiled for proteomes using the Human Angiogenesis Protein Array. (**B**) Levels of PDGF-BB expression were quantified in each protein array sample. (**C**) EPCs were incubated with VEGF (100 ng/mL) and melatonin (0.1-1 mM) for 24 h, and PDGF-BB expression was examined by western blot analysis (*n* = 3). (**D**,**E**) EPCs were transfected with PDGF-BB cDNA overnight, then left untreated or were treated with VEGF (100 ng/mL) and melatonin (1 mM) for 24 h, before being examined by the Transwell (*n* = 3) and tube formation assays (*n* = 3). (**F**) Levels of PDGF-BB expression were analyzed in gene expression profile records downloaded from the eyeIntegration v1.0 database for normal retina tissue samples and neovascular AMD tissue samples. * *p* < 0.05 compared with the control group; # *p* < 0.05 compared with the VEGF-treated group.

**Figure 5 cells-12-00799-f005:**
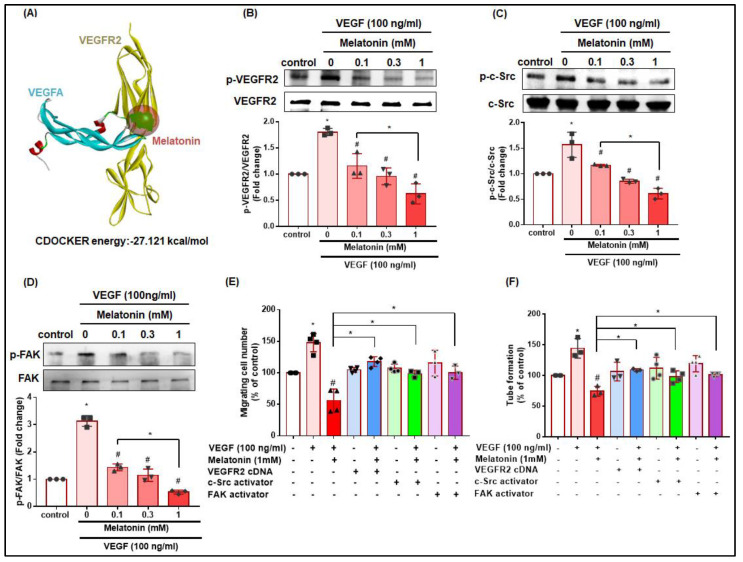
Melatonin reduces the VEGF-induced stimulation of EPC migration and tube formation by inhibiting the VEGFR2/c-Src/FAK signaling pathway. (**A**) Molecular docking software revealed a binding affinity between melatonin and VEGFR2. (**B**–**D**) EPCs were incubated with VEGF (100 ng/mL) and different concentrations of melatonin (0.1–1 mM) for 2 h. VEGFR-2, c-Src and FAK phosphorylation was examined by western blot (*n* = 3). (**E**,**F**) EPCs were transfected with VEGFR2 cDNA or treated with FAK or c-Src activators overnight, then left untreated or were treated with VEGF (100 ng/mL) alone and in combination with melatonin (1 mM) for 24 h, before being examined by the Transwell (*n* = 4) and tube formation assays (*n* = 4). * *p* < 0.05 compared with the control group; # *p* < 0.05 compared with the VEGF-treated group.

**Figure 6 cells-12-00799-f006:**
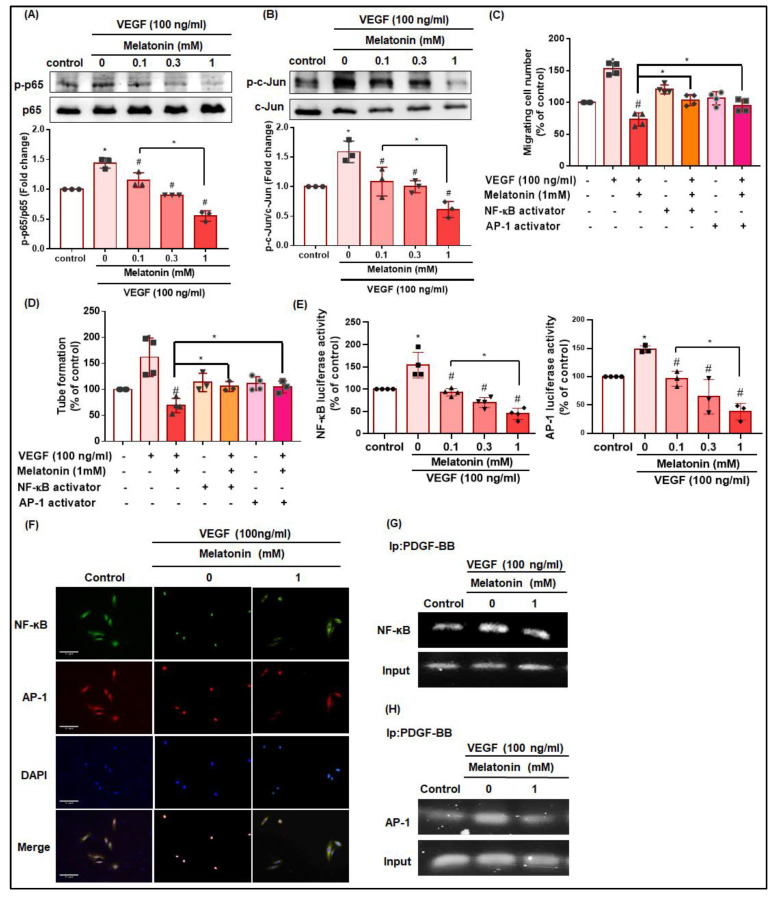
Melatonin inhibits the VEGF-induced stimulation of EPC angiogenesis by inhibiting NF-κB and AP-1 activation. (**A**,**B**) EPCs were incubated with VEGF (100 ng/mL) and different concentrations of melatonin (0.1–1 mM) for 2 h. p65 and c-Jun phosphorylation was examined by western blot (*n* = 3). (**C**,**D**) EPCs were treated with NF-κB or AP-1 activators overnight, then left untreated or were treated with VEGF (100 ng/mL) alone and in combination with melatonin (1 mM) for 24 h, before being examined by the Transwell (*n* = 4) and tube formation assays (*n* = 4). (**E**) EPCs were transfected with the NF-κB or AP-1 luciferase plasmids and then treated with VEGF (100 ng/mL) and different concentrations of melatonin (0.1–1 mM) before determining luciferase activity (*n* = 4). EPCs were then treated with VEGF (100 ng/mL) and melatonin (1 mM), before undergoing (**F**) immunofluorescence staining with NF-κB and AP-1 antibodies, or (**G**,**H**) a ChIP assay (*n* = 3). * *p* < 0.05 compared with the control group; # *p* < 0.05 compared with the VEGF-treated group.

**Figure 7 cells-12-00799-f007:**
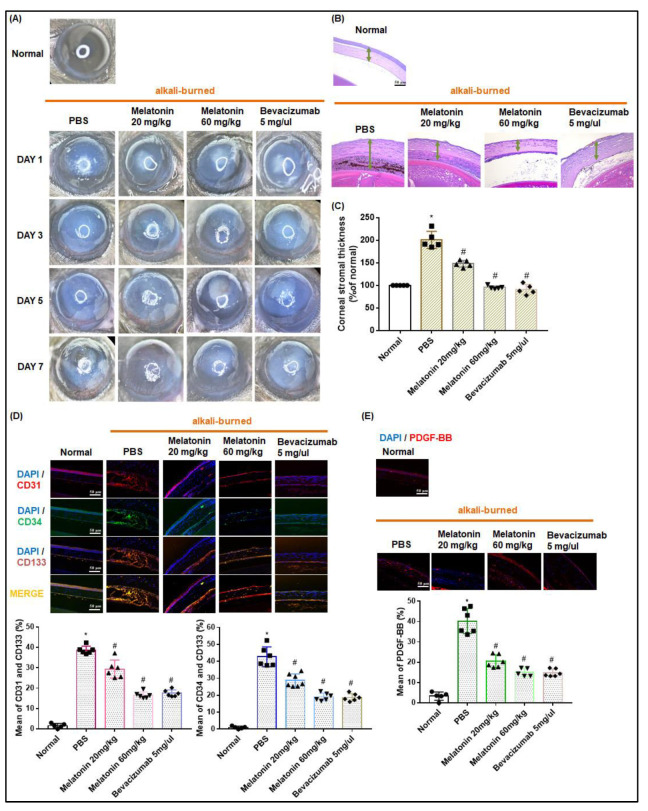
Antiangiogenic effects of melatonin in the corneal alkali burn model. (**A**) Photos of a normal cornea and an alkali-burned cornea. Stereomicroscopic findings for eyes from mice (*n* = 6 per group) on days 1, 3, 5, and 7 after treatment with PBS, melatonin (20 mg/kg or 60 mg/kg), or bevacizumab (5 mg/μL). (**B**,**C**) At 7 days after the alkali burn injury, corneal stromal thickness was detected by H&E staining and quantified in mice with uninjured corneas, mice with untreated injured corneas, melatonin-treated mice, or bevacizumab-treated mice. (**D**,**E**) Levels of CD31, CD34, CD133 and PDGF-BB expression in corneas were subjected to co-immunofluorescence staining and quantified. * *p* < 0.05 compared with uninjured corneas; # *p* < 0.05 compared with damaged corneas.

**Figure 8 cells-12-00799-f008:**
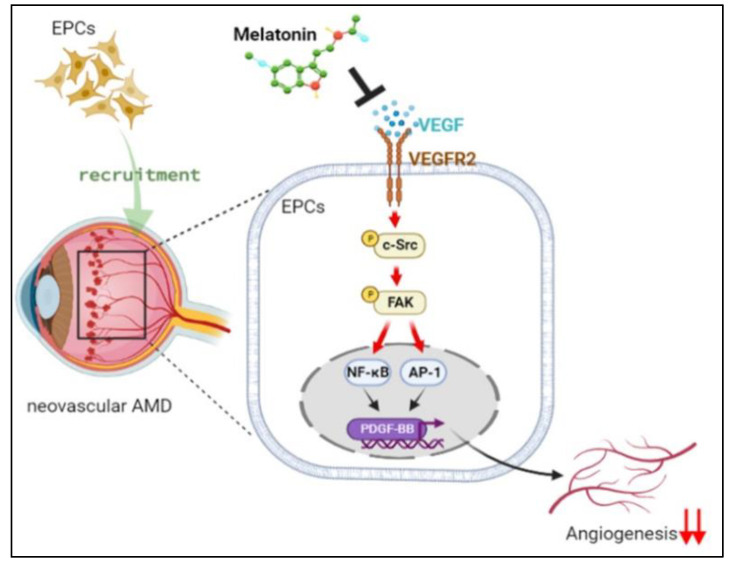
The schematic diagram summarizes the proposed mechanism whereby melatonin suppresses VEGF-induced EPC angiogenesis in neovascular AMD. Melatonin suppresses VEGF-induced increases in the production of PDGF-BB, the recruitment of EPCs, and EPC angiogenesis by inhibiting VEGFR2, c-Src, FAK, NF-κB and AP-1 signaling.

## Data Availability

The data generated and analyzed will be made from the corresponding author on reasonable request.

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
