# Peer review of "Melatonin Inhibits VEGF-Induced Endothelial Progenitor Cell Angiogenesis in Neovascular Age-Related Macular Degeneration"

_cells, 2023, doi:10.3390/cells12050799_

Round 1

Reviewer 1 Report

The article titled "Melatonin inhibits VEGF-induced endothelial progenitor cell angiogenesis in neovascular age-related macular degeneration" is well-written. 

However, I have a few comments about the figures,

1. Figure 2C the pictures are not clear, and challenging to see any effect.

2. Figure 3C Can you show figures with higher magnification so it's clearer?

Author Response

  1. Figure 2C the pictures are not clear, and challenging to see any effect.

A: Clear pictures have now been provided (Fig. 2C).

  1. Figure 3C Can you show figures with higher magnification so it's clearer?

A: Pictures with higher magnification have now been provided (Fig. 3C).

Reviewer 2 Report

The article ”Melatonin inhibits VEGF-induced endothelial progenitor cell angiogenesis in neovascular age-related macular degeneration” responds to the interest if melatonin can be used as a treatment for reducing endothelial progenitor cell angiogenesis in neovascular age-related macular degeneration. The scientific demonstration is well conducted. The improvements can involve:

1. Promoting or inhibiting angiogenesis is part of the homeostatic balance with positive and negative effects outside the optimum range. Melatonin can influence this balance, so for the entire organism, can we have a better picture of using this hormone for treatment purposes?

2. Is there a cost-effectiveness approach to compare using melatonin or other existing anti-VEGF agents approved by FDA, for example, bevacizumab, aflibercept, or ranibizumab?

3. The introduction can exploit the role of melatonin in retinal physiology and connections with the light/dark cycle. The paragraph between lines 82-94 can be sustained with more data, and the scientific construction can be better elaborated and developed.

Author Response

  1. Promoting or inhibiting angiogenesis is part of the homeostatic balance with positive and negative effects outside the optimum range. Melatonin can influence this balance, so for the entire organism, can we have a better picture of using this hormone for treatment purposes?

A: This theme has accordingly been addressed in the Discussion section, as follows:

“The promotion or inhibition of angiogenesis is part of the homeostatic balance, with positive and negative effects outside the optimum range. Melatonin influences this balance, with evidence from several clinical research investigations demonstrating that this hormone has antiangiogenic effects in cancer and chronic ocular diseases. For instance, in cancer treatment, adjuvant melatonin appears to be very effective in early-stage disease and helps to reduce the side effect profiles after radiotherapy and chemotherapy. In patients with central serous chorioretinopathy, treatment with oral melatonin (3 mg) three times daily for 1 month resulted in significant improvements from baseline in best-corrected visual acuity (BCVA) and decreases in central macular thickness (CMT), without any adverse effects. The attractive side effects profile and relatively low cost of melatonin suggest that this hormone may be appropriate in a chronic ocular disease such as AMD, although supporting data from large prospective studies are needed before melatonin can be used in the clinic.” (Lines 427-439)

  1. Is there a cost-effectiveness approach to compare using melatonin or other existing anti-VEGF agents approved by FDA, for example, bevacizumab, aflibercept, or ranibizumab?

A: We have now added more detail to the Discussion text about the cost-effectiveness of anti-VEGF therapy and melatonin, as follows:

“The attractive side effects profile and relatively low cost of melatonin suggest that this hormone may be appropriate in a chronic ocular disease such as AMD, although supporting data from large prospective studies are needed before melatonin can be used in the clinic. The FDA-approved anti-VEGF agents bevacizumab, aflibercept, and ranibizumab, have shown good therapeutic results in neovascular AMD, but anti-VEGF agents are limited by the necessity for monthly injections in the clinic and the long-term nature of treatment. Moreover, the high cost of anti-VEGF therapy is a heavy burden for patients.” (Lines 436-443)

  1. The introduction can exploit the role of melatonin in retinal physiology and connections with the light/dark cycle. The paragraph between lines 82-94 can be sustained with more data, and the scientific construction can be better elaborated and developed.

A: More data have now been provided in the Introduction section detailing the role of melatonin in retinal physiology and connections with the light/dark cycle, as follows:

“The synthesis and release of the neurohormone melatonin enables organisms to respond to circadian and seasonal rhythms. Melatonin is mainly secreted by the pineal gland under the influence of light stimulation of the retina; to a lesser extent, melatonin is also synthesized within the eye, where melatonin uses ocular structures to mediate a variety of diurnal rhythms and physiological processes within the eye. Increasing the concentration of melatonin at night promotes sleep, while decreasing the concentration of melatonin during the day promotes alertness. Melatonin concentrations tend to decrease with age. The effects of melatonin are beneficial for numerous physiological functions, including the promotion of ocular surface wound healing, reductions in inflammation and oxidative stress, and angiogenesis.” (Lines 82-91)

Reviewer 3 Report

The study entitled ‘Melatonin inhibits VEGF-induced endothelial progenitor cell angiogenesis in neovascular age-related macular degeneration’ is very interesting and demanding in anti-angiogenic therapy associated with neovascular AMD. Overall, the study design is transparent, and the authors employed several in vitro and in vivo approaches to consider melatonin as a potential drug to treat VEGF-induced excess angiogenesis. However, I have several concerns about the current form of the manuscript.

Ø  In the materials and methods section, the authors mentioned that they used the student’s t-test. However, in most figures, there are more than two groups to compare. Therefore, it is required to perform appropriate statistical analysis, such as ANOVA followed by a post hoc test for multiple comparisons.

Ø  In figure 1, there is no explanation about the groups indicated by the X-axis. The authors need to introduce the groups in the results section (Subheading 3.1) as well as in the figure 1 legend.

Ø  Is there any statistically significant difference between the 0.1 mM and 1 mM melatonin treatments in tube formation, migrating cell numbers, and all other parameters used throughout the manuscript? If there are differences, then the author should update all the figures with those significant values. 

Ø  In figure 2, panel D: The micrographs that represent VEGF and 0.1 mM Melatonin+VEGF are not consistent with their quantifications. It seems to me that both micrographs have a similar number of branches.

Ø  All figure legends are missing the number of biological replicates value (n=#?).

Ø  The authors should provide representative micrographs for all the figures (4, 5 & 6) where they performed the tube formation assay.

Ø  Since angiotensin II has numerous off-target effects using FAK-specific siRNA/shRNA or specific drugs can concrete the involvement of FAK in the VEGFR2/c-Src/FAK signaling pathway.

Ø  There are several grammatical mistakes throughout the whole manuscript.

Author Response

  1. In the materials and methods section, the authors mentioned that they used the student’s t-test. However, in most figures, there are more than two groups to compare. Therefore, it is required to perform appropriate statistical analysis, such as ANOVA followed by a post hoc test for multiple comparisons.

A: ANOVA followed by post hoc testing was used for the analysis of multiple comparisons. This information has been added into the Methods section, as follows:

 “One-way analysis of variance (ANOVA) followed by post hoc testing was used for statistical analyses of multiple group comparisons.” (Lines 196-197)

  1. In figure 1, there is no explanation about the groups indicated by the X-axis. The authors need to introduce the groups in the results section (Subheading 3.1) as well as in the figure 1 legend.

A: More explanatory information has been added accordingly, as follows:

“EPC-specific markers CD34 and CD133 and blood vessel markers CD31 and VEGF were all highly expressed in AMD retinas and graded with Minnesota Grading System (MGS) scores of 1 to 4” (Lines 204-207; 222-223)

  1. Is there any statistically significant difference between the 0.1 mM and 1 mM melatonin treatments in tube formation, migrating cell numbers, and all other parameters used throughout the manuscript? If there are differences, then the author should update all the figures with those significant values. 

A: We thank the Reviewer for this suggestion. The statistically significant difference between the 0.1 mM and 1 mM melatonin concentrations has now been added.

  1. In figure 2, panel D: The micrographs that represent VEGF and 0.1 mM Melatonin+VEGF are not consistent with their quantifications. It seems to me that both micrographs have a similar number of branches.

A: The micrographs in Fig. 2D representing VEGF and 0.1 mM Melatonin+VEGF have now been changed with images that more clearly show the quantifications (Fig. 2D).

  1. All figure legends are missing the number of biological replicates value (n=#?).

A: The n value has now been added to all figure legends.

  1. The authors should provide representative micrographs for all the figures (4, 5 & 6) where they performed the tube formation assay.

A: Representative tube formation micrographs for Figures 4, 5 & 6 have now been provided in the Supplementary file (Fig. S1, S2 & S3, respectively).

  1. Since angiotensin II has numerous off-target effects using FAK-specific siRNA/shRNA or specific drugs can concrete the involvement of FAK in the VEGFR2/c-Src/FAK signaling pathway.

A: We used a FAK activator (angiotensin II; sc-363643) (Line 108) to reverse melatonin-induced inhibition of EPC migration and tube formation (Fig. 5E&F). As yet, no other more specific FAK activators have been developed for experimental use. FAK siRNAs cannot be used to confirm angiotensin II activity and the involvement of FAK in the VEGFR2/c-Src/FAK signaling pathway.

  1. There are several grammatical mistakes throughout the whole manuscript.

A: The manuscript has been revisited by a native English-language speaker with medical editing experience.

Round 2

Reviewer 2 Report

The authors responded to all my previous improvement comments, and now my recommendation is for acceptance in the present form.